

# Research on fault location algorithm of TPSS based on PSOA

Yunqian Jia

Shanghai Institute of Technology, School of Railway Transportion, Shanghai, China

## ABSTRACT

It is extremely important to research traction power supply system (TPSS) protection technology in order to ensure the safe operation of urban rail transit. A TPSS includes rails, return cables, rail potential limiting devices, one-way conducting devices, drainage cabinets, ballast beds, and tunnel structural reinforcements. In urban rail transit, on the basis of the dynamic characteristics of the TPSS, a fault location algorithm based on particle swarm optimization algorithm (PSOA) is developed. An evaluation of multi-point monitoring data is proposed based on fuzzy processing of the average value of polarization potential forward deviation and multi-attribute decision-making. Monitoring points and standard comparison threshold values are determined by the distribution law of stray currents. In conjunction with the actual project, the model is trained using field measured data. Based on the results, TPSSOA is able to achieve optimal discharge current control, reduce network losses and improve power quality. Moreover, the reconstruction results demonstrate the high usability of the proposed method, which will provide guidance to design the TPSS in the future.

## INTRODUCTION

Urban development has been restricted by the bottleneck problem of urban internal traffic development as the economy and urbanization have grown rapidly. Due to its advantages of large traffic volumes, low energy consumption, safety, fast and punctual comfort, large and medium-sized cities now rely on urban rail transit to alleviate traffic congestion. Nearly 120 cities around the world have built urban rail transit systems within the past 100 years, and the operating mileage of these systems exceeds 7,000 km, according to incomplete statistics. Electric traction system can be divided into DC electric traction and AC electric traction. A traction power supply system (TPSS) requires constant starting torque and good speed regulation characteristics. Due to the volume and cost of high-power converter, AC electric traction is not suitable to be installed separately in each traction locomotive. However, a DC series excitation motor can better meet these requirements, so the DC traction system is mostly adopted (*He et al., 2019*; *Liu et al., 2020*). The DC traction system is widely used in urban rail transit system because it has the following three characteristics: (1) An urban rail transit system (underground tunnel or light rail) mostly runs through the urban area, and the corridor space of electric traction line is limited. (2) As a traction load, the locomotive has the characteristics of high density,

Corresponding author
Yunqian Jia,
216151112@mail.sit.edu.cn

short running distance and frequent changes of running state. Locomotive operation can be divided into three states: starting, idling and braking. Under different states, the system current changes greatly with high frequency. The traction current is large when starting, small when idling, and reverse when braking. (3) In order to increase transport capacity, on-board equipment should be simplified as far as possible (*Zhong et al., 2020*).

As an important part of TPSS, the reflux system and its influence have been paid more attention in the world. A gratifying progress is being made on DC traction power supply return systems due to the rapid development of urban rail transit. As much as possible, rail leakage current should be reduced to eliminate stray current from the source. There is still widespread use of many protection principles and measures summarized by *Yang et al. (2018)*. Among the specific protection measures outlined by *Chen et al. (2020)* are: maintaining a high level of rail ground insulation. Adopt effective grounding and connection scheme. The upper and lower rails are connected by welding. The stray current collection system should be reasonably established in order to reduce the resistance of the return rail and to reduce the distance between traction posts. Track and structural steel are isolated, especially when ballast bed drainage network is used to ensure that the track is dry and the drainage is timely. The necessity of monitoring stray current leakage outside the system is proposed. In addition, high rail potential will cause a lot of stray current leakage, the comprehensive measures discussed by *Chen et al. (2020)* included increasing the voltage on the traction network, increasing the cross-sectional area of return rails, welding long rails, and reducing the distance between substations. Due to the short circuit between the track and the ground, the original insulation protection measures designed on the line will lose the protection function. Taking effective measures for active treatment is therefore necessary to address the phenomenon.

Many cities have the above problems with their subway operations, and the relevant protection technology needs to be improved and solved. It is extremely important to research TPSS protection technology in order to ensure the safe operation of urban rail transit. On the basis of the dynamic characteristics of the TPSS, a TPSSOA algorithm was developed. The exhaust flow control is equivalent to duty cycle adjustment based on the dynamic flow drainage method of multi-point monitoring data. The predictive model can effectively control discharges after being trained on field data, and using the actual project to implement drainage control.

## RELATED WORKS

*Chen et al. (2020)* analyzed the characteristics of DC TPSS. Based on the assumption that the rail ground insulation is uniformly distributed, it was established that rail resistance and rail ground transition resistance are variables in a continuous distribution model and relevant formulas are derived and analyzed. Considering the influence of ballast reinforcement structure, the continuous distribution model of rail, structure and earth was established. According to *Lin, Feng & Sun (2019)*, the simplified boundary conditions were adopted in the theoretical analysis, because the parameters were assumed to have a continuous distribution in the actual situation. The theoretical calculation and simulation results obtained could not completely correspond to the actual situation. Therefore, the

various analysis results at this stage can only be used as the reference for qualitative analysis, and there is a big gap between the accurate calculation of the distribution and influence of each parameter. As a result of their work, *Dai et al. (2021)* have developed the discrete distribution model, given the software program based on one car and one section, and conducted simulation analyses. They have also calculated the effect of subway stray current using finite element method, which is the basis for determining the protection range of stray current and quantifying the influence degree of subway stray current on surrounding buried metal structures. The traditional circuit theory has encountered many difficulties in the analysis of reflux system. Many scholars try to use other theoretical models to analyze the reflux system. *Nezevak, Cheremisin & Shatokhin (2018)* developed a model of multi locomotive and multi section using the node equation matrix method, conducted simulation analysis, and verified the simulation results on the ground. Based on the electric field theory, hemispherical electrode was used to solve the electric field distribution under the influence of stray current, so as to calculate the leakage current in buried metal and predict the corrosion of stray current to buried metal. *Zhang et al. (2018)* derived analytical formulas for the track voltage, track current, stray current, rail ground transition resistance, and other parameters under the continuous and discrete methods, and carried out simulation analyses.

On the basis of a section of track, *Yuan et al. (2019)* derived the relationship between rail ground voltage and stray current. When there is a buried metal structure near the track, the potential gradient of leakage current is calculated by using field theory. When the parallel or crossed buried metal is in the electric field, there will be potential difference and stray current corrosion. The limitations of practical application such as discontinuous transition resistance and uneven soil resistivity are discussed. According to *Lin, Chen & Wang (2018)*, stray current distribution in DC traction systems is characterized by time variations. As a return path, *Wang et al. (2020)* considered a rail as a ferromagnetic conductor with an irregular cross-section. When the traction current changes sharply, it is necessary to analyze the influence of its transient characteristics on the rail potential. Taking the independent rail as an example, the frequency-domain transient characteristic parameters are calculated and the transient parameter model is established. The influence of frequently changing traction current on rail transient parameters is discussed.

## TPSSOA SCHEME

### PSOA

Based on research on bird predation behavior, Kennedy and Eberharty developed PSOA. Birds cannot know the exact location of food if there is only one piece in their foraging range, but they can feel the distance between themselves and the food when foraging in a flock. Therefore, if you want to spend the least time to find food, the birds can follow the nearest bird to find food. In the PSOA, the bird swarm is the particle swarm, and a single bird corresponds to a single particle. The process of PSOA solving the optimization problem is accompanied by the updating of particle velocity and position. In the process of algorithm optimization, each particle has its own speed and position, and updates the speed and position independently.

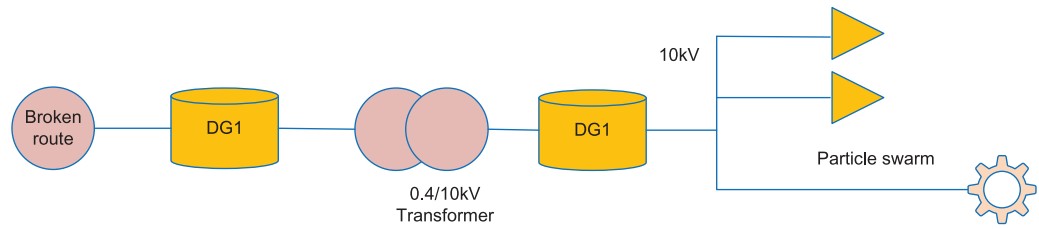

**Figure 1 DG schematic diagram.** The schematic diagram of DG in orbit is shown.

At the same time, the particles will constantly adjust the search direction according to their own experience and the experience of other particles, and finally realize the search of the optimal value.

The schematic diagram of DG in orbit is shown in Fig. 1. As a final conclusion, the population is positioned optimally depending on the optimization problem. The updating formulas of velocity and position are shown in Eqs. (1) and (2).

$$X_{id}^{k+1} = X_{id}^k + V_{id}^{k+1} X_{i+1} = r_1 k (1 - V_{id})(V_{id} - k + 1) \tag{1}$$
$$V_{id}^{k+1} = w V_{id}^k + c_1 r_1 (P_{id} - X_{id}) \tag{2}$$

In the present optimal value formula, $X_{id}^{k+1}$ for the first time $k + 1$, the velocity and position of the second iteration, $w$. Is the inertia weight, $V_{id}, X_{id}$. The velocity and position of the second iteration; $c_1$, $c_2$ are the acceleration factor, which is usually greater than 0; $r_1, r_2$ are interval $[0, 1]$. The random constant in the $P_{id}, X_{id}$ are the optimal positions of individuals and populations respectively. Many scholars have made efforts to improve IPSO, and put forward an improved method to make the inertia weight decrease linearly. In order to improve the algorithm's search ability, the compression factor is introduced, and the inertia weight strategy of nonlinear decreasing is introduced. The above-mentioned improvement measures can solve simple optimization problems, but when the problems are complicated, the optimization effect will be greatly reduced. To solve this problem, this article studies the learning factors in PSO $c_1$, $c_2$. And inertia weight $w$. Optimization is performed on three parameters that significantly influence algorithm performance. The improved inertia weight is shown in Formula (3).

$$c_1 = c_{1\max} + c_{1\min} \sin(1 - t)\pi \tag{3}$$

$c_1$ is the current fitness value, $c_{1max}$ is the minimum fitness value, $c_{1min}$ is the average fitness value. It represents the number of iterations that have been carried out so far and the maximum number of iterations that can be carried out. In this way, with the continuous iteration of the algorithm, $w$. In the iterative process, the value of the fitness value is dynamically adjusted to improve its optimization performance. In addition, this article also adopts the strategy of dynamically adjusting the acceleration factor based on the change of fitness value $t$. It needs to be mapped to chaotic sequences $Z_i$. The mapping process is shown in Eq. (4).

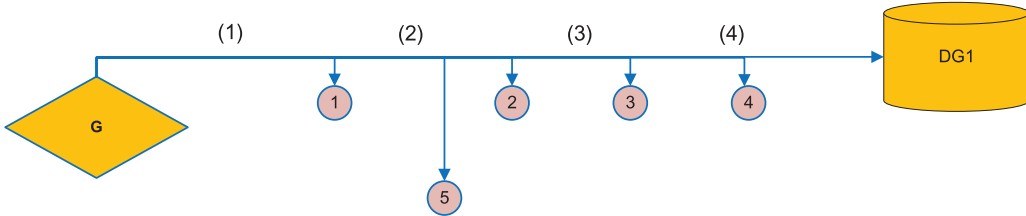

**Figure 2 Schematic diagram of power supply network.** It is commonly used to use integer values of 0–0 as a parameter in urban rail transit models. The simple power supply network is shown. Taking this as an example, because the network only has the main power supply for power supply, the current direction is that the system power supply flows to the load. When the fault occurs in the section, the traction network corresponding to the section monitors the fault current passing through, Thus, for the switch with fault over-current, the status value of the corresponding node is "1", whereas for the switch without fault over-current, the status value is "0". That is to say, for a single switch node, only the downstream section fault needs to be considered.

$$Z_i = Y_i - \frac{Y_{min}}{Y_{imax} - Y_{imin}} \tag{4}$$

Final chaotic variable $Y_i$ is required for the generation process, as shown in Eq. (5).

$$Y_i = Z_i(Y_{imax} - Y_{imin}) + Y_{imin} \tag{5}$$

### Specific scheme of TPSSOA

It is commonly used to use integer values of 0–0 as a parameter in urban rail transit models. The simple power supply network is shown in Fig. 2.

Taking this as an example, because the network only has the main power supply for power supply, the current direction is that the system power supply flows to the load. When the fault occurs in the section, the traction network corresponding to the section monitors the fault current passing through, Thus, for the switch with fault over-current, the status value of the corresponding node is "1", whereas for the switch without fault over-current, the status value is "0". That is to say, for a single switch node, only the downstream section fault needs to be considered.

1. TPSSOA firstly collects and uploads the fault current information in real time by using traction network, and realizes the conversion from fault current information to fault vector by using improved fault integer programming model, that is, according to the actual situation of fault current direction detected by traction network, it can be converted into "1", "−1" or "0".

2. Then the switch function of urban rail transit with rail transit is constructed, and the equivalent fault vector of fault location is calculated. The equivalent fault vector is input into the algorithm, the population dimension of PSO and the coding length of genetic algorithm are set as the total number of feeder sections, and the evaluation function is the objective function of the algorithm.

3. Finally, the optimal particle position of PSO is output to realize the fault section location. In this article, an adaptive PSO is proposed by dynamically improving the

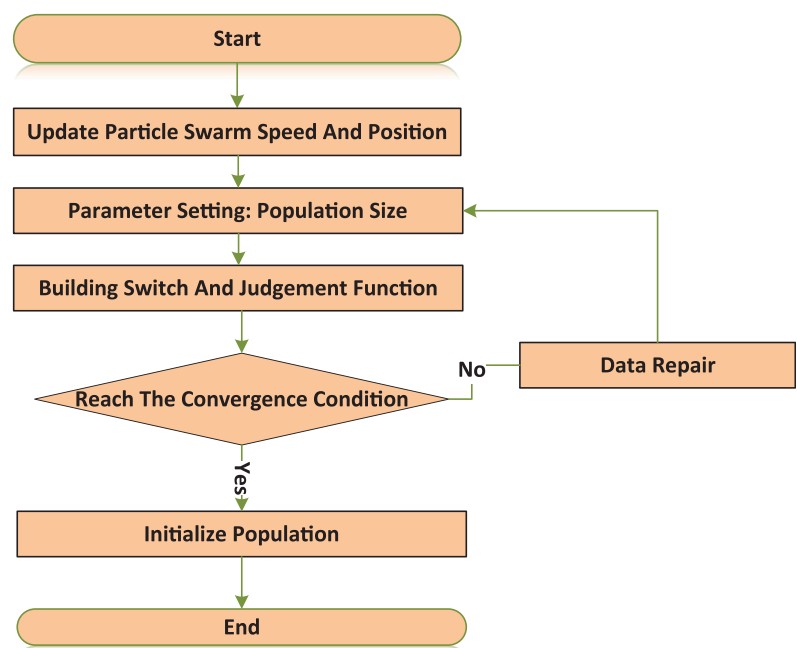

**Figure 3** **TPSSOA flow chart.** TPSSOA firstly collects and uploads the fault current information in real time by using traction network, and realizes the conversion from fault current information to fault vector by using improved fault integer programming model, that is, according to the actual situation of fault current direction detected by traction network, it can be converted into "1", "−1" or "0". Then the switch function of urban rail transit with rail transit is constructed, and the equivalent fault vector of fault location is calculated. The equivalent fault vector is input into the algorithm, the population dimension of PSO and the coding length of genetic algorithm are set as the total number of feeder sections, and the evaluation function is the objective function of the algorithm. Finally, the optimal particle position of PSO is output to realize the fault section location. In this article, an adaptive PSO is proposed by dynamically improving the fitness value of the PSOA. Combined with the adaptive genetic algorithm, the search iteration ability of the adaptive genetic PSOA is proposed. The corresponding flow chart is shown.

fitness value of the PSOA. Combined with the adaptive genetic algorithm, the search iteration ability of the adaptive genetic PSOA is proposed.

The corresponding flow chart is shown in Fig. 3. The specific steps of fault location using the improved algorithm are as follows:

1. There were set parameters such as the number of iterations and the size of the population.
2. Initiate the population information and construct the switch function and evaluation function.
3. Calculate the fitness function value and perform the genetic algorithm selection.
4. We calculate the crossover probability of genetic algorithms and perform crossover operations.

5. The mutation probability of genetic algorithm is calculated and mutation operation is performed.
6. Judge whether the current fitness value meets the convergence condition, and if so, execute the next step. Otherwise, return to step (3).
7. Output population information and initialize the velocity and position of particles.
8. The current fitness value is solved and the velocity and position of particles are updated.
9. In the case of a convergence condition being met, execute the next operation if it has been met.
10. The optimal particle output is the fault section.

When TPSSOA determines a fault point and its opposite bus, it compares the reflected waves by polarity. Wavelet analysis is used to identify and extract the fault signal, and the structure and operation characteristics of the metro DC traction power supply system are different from those of high-voltage transmission lines. In general, there are two incoming lines and four feeders connected with the up and down lines in the left and right sections respectively. Moreover, the change of traction load will also bring interference to the direction discrimination of traveling wave. If the discrimination is wrong, it will cause great ranging error. Therefore, TPSSOA combines with fault analysis method, calculates the fault location by the optimized fault location algorithm proposed above, and when the reflection waves of a fault point and the opposite bus arrive at the measuring end, that is the time interval when the reflected wave of the fault point arrives. In the area, the accurate arrival time of fault point reflection wave and opposite bus reflection wave is determined respectively, and then the fault location is carried out by substituting it into the formula. Among them, the fault analysis method is applicable to any section of data in the whole instantaneous process, which requires a small amount of data, and can share the data source with the traveling wave method without additional workload in data acquisition.

## MODEL SIMULATION AND RESULT ANALYSIS

The reference voltage of rail transit traction system used in this experiment is 18 kvA, the reference power is 10 mvA, and the network load is 3715 + j2300 kvA. The parameters of the TPSSOA algorithm is shown in Table 1.

When DG is not connected, when there is no DG in urban rail transit, the status values of the switch are only "0" and "1". If there is high temperature, humidity and other harsh conditions, the fault information collected may be different from the actual value. Therefore, the information distortion is considered in this experiment. The simulation results are shown in Table 2.

It can be seen from the results in Table 2 that the fault area can also be accurately located when multiple sections are in fault at the same time and the fault information is not accurate. At the same time, the fault section can be accurately determined even if the fault information of multiple feeder sections is not accurate. As a result of the simulations, the fault location method proposed can dynamically adapt to the flow of DGs into and out of the fault, and the experimental results are not disturbed by distorted information. In order

**Table 1 Parameters of the TPSSOA algorithm.** The reference voltage of rail transit traction system used in this experiment is 18 kvA, the reference power is 10 mvA, and the network load is 3715 + j2300 kvA. The parameters of TPSSOA algorithm is shown.

| Parameter | PSOA | DG | Testing sample |
|---|---|---|---|
| Population | 45 | 51 | 35 |
| Max iterations | 45 | 51 | 35 |
| Dimension | 61 | 23 | / |
| Code length | 76 | 23 | 25 |

**Table 2 DG access results.** When DG is not connected, when there is no DG in urban rail transit, the status values of the switch are only "0" and "1". If there is high temperature, humidity and other harsh conditions, the fault information collected may be different from the actual value. Therefore, the information distortion is considered in this experiment. The simulation results are shown. It can be seen from the results that the fault area can also be accurately located when multiple sections are in fault at the same time and the fault information is not accurate. At the same time, the fault section can be accurately determined even if the fault information of multiple feeder sections is not accurate. As a result of the simulations, the fault location method proposed can dynamically adapt to the flow of DGs into and out of the fault, and the experimental results are not disturbed by distorted information.

| Fault | Equivalent fault vector | Output optimal solution | Node |
|---|---|---|---|
| Single fault | [1100110100011001010001100] | [1010110100011001011110000] | (1) |
| Multiple faults | [1000110100011001011111101] | [1110110100011001011111000] | (3) |
| Multiple information distortion | [1110110100011001011111101] | [1000110100011001011111100] | (2) |

to reflect the advantages of the improved algorithm in fault location, this article compares the TPSSOA algorithm with the improved PSOA (IPSO) after introducing the compression factor and linear decreasing inertia weight. The formula of linear decreasing inertia weight and compression factor is shown in Formula (6).

$$w = w_{max} - (w_{max} - w_{min})T_{max} \tag{6}$$

where, $w_{max}$ is the compression factor and $T_{max} = C_1$. Different fault conditions are set, and simulation and comparative analysis are carried out in terms of rapidity. DG grid connection coefficient is set as $[k1, k2, k3] = [11000110111110]$.

The iterative curves of the two algorithms in fault location are shown in Fig. 4 (No distortion in single section) and Fig. 5 (no distortion in multiple sections). The horizontal axis is the number of iterations and the vertical axis is the fitness value. The default fault location is feeder section (1), and the fault information has no distortion.

It can be seen from the results in Fig. 4 that both algorithms can achieve minimum values within the maximum number of iterations, but the speed of TPSSOA is the faster than IPSO. The preset fault location is feeder section (2), and the fault information at switch (3) is distorted. It can be seen from Fig. 5 that IPSO falls into a local optimal value at the early stage of iteration. In order to obtain the global optimal value, more iterations are needed. However, TPSSOA achieves global extremum with less iterations. The comparison results show that IPSO algorithm can hardly locate multiple section faults with a large
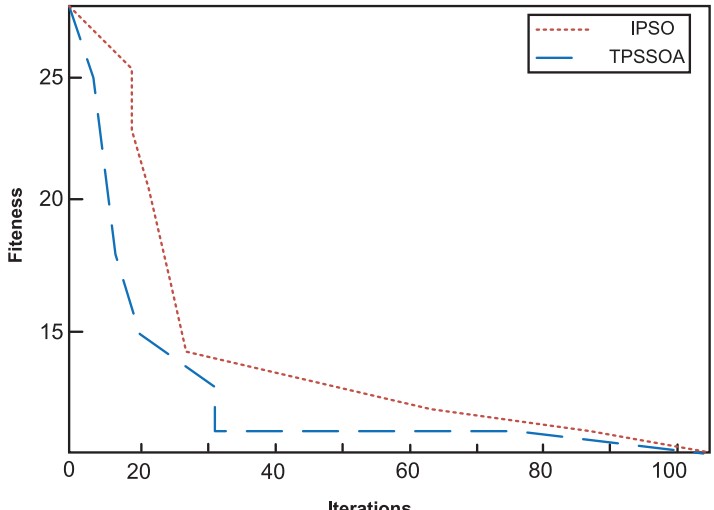

**Figure 4  No distortion in single section.** The iterative curves of the two algorithms in fault location are shown. The horizontal axis is the number of iterations, and the vertical axis is the fitness value. The default fault location is the feeder segment (1), and the fault information is not distorted. It can be seen from the results that both algorithms can achieve minimum values within the maximum number of iterations, but the speed of TPSSOA is the fastest and IPSO is the slowest. The preset fault location is feeder section (2), and the fault information at switch (3) is distorted.

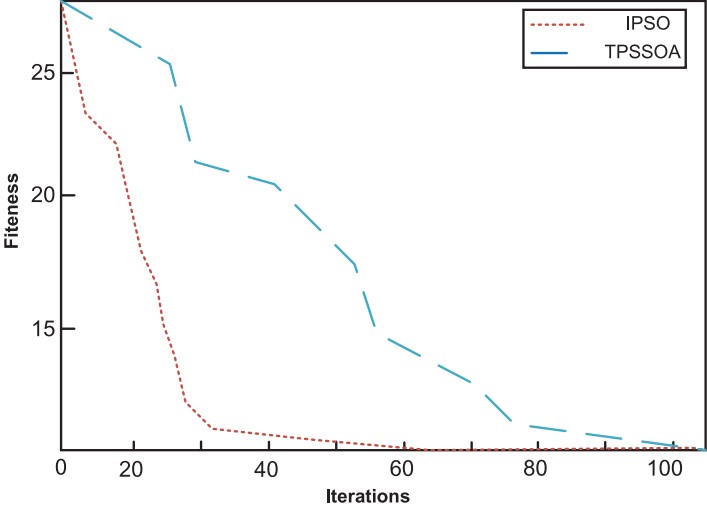

**Figure 5  No distortion in multiple sections.** The iterative curves of the two algorithms in fault location are shown. The horizontal axis is the number of iterations, and the vertical axis is the fitness value. The default fault location is the feeder segment (1), and the fault information is not distorted. It can be seen that IPSO falls into a local optimal value at the early stage of iteration. In order to obtain the global optimal value, more iterations are needed. However, TPSSOA achieves global extremum with less iterations.

amount of inaccurate information, and the accuracy of single fault location is not high, so it is only suitable for simple urban rail transit. The TPSSOA algorithm has high positioning accuracy and is hardly affected by the increase of fault section and distortion information. In this article, fault reconstruction experiments are used to demonstrate the effectiveness of

the method proposed. On urban rail transit without grid connected DG and urban rail transit with different types of DG connected. Reconstruction of the fault leads to the restoration of power supply in the non-fault area and a reduction of network losses. Compared with IPSO algorithm, TPSSOA uses the improved algorithm to reconstruct the switch combination network with lower loss and less time-consuming. In addition, voltage amplitudes and voltage distributions are improving. At the same time, after fault reconstruction, the combination of (3) (4) (5) is changed to (1) (2) (3) (5), the network loss is reduced from 111.6606 to 72.4095 kw, and the minimum node voltage is also increased from 0.9342 to 0.9599. Compared with the fault reconstruction scheme using IPSO algorithm, the reconstruction scheme using the TPSSOA algorithm can reduce more network loss and increase more voltage amplitude. In conclusion, IPSO is easy to be disturbed by distorted information when using the above algorithm, and it falls into local extremum many times in the whole iteration process, so it is only suitable for fault location of simple urban rail transit. The TPSSOA proposed in this article has a strong anti-interference ability for distortion information, and can quickly obtain global extreme value, so as to make accurate identification of fault section. It has excellent fault tolerance and stability, and greatly improves the search ability of the algorithm. It can be used in urban rail transit with DG.

## CONCLUSION

TPSS in urban rail transit are constantly evolving and improving, enhancing reliability and operation economy, but also increasing the probability of failure, and network architectures are becoming more complex. Due to limitations in existing fault location and recovery reconstruction methods, this article presents an algorithm based on PSO that improves the location model, optimizes the PSOA, and enables the algorithm to be more efficient in optimizing the location model. An improved fault location method is developed, and DG grid connection and distortion information are not affecting the experimental results of the fault location experiment. We conclude that the proposed fault location method can reduce network losses and improve power quality based on the reconstruction results.

### Funding
The author received no funding for this work.

### Competing Interests
The author declare that they have no competing interests.

### Author Contributions
- Yunqian Jia conceived and designed the experiments, performed the experiments, analyzed the data, performed the computation work, prepared figures and/or tables, authored or reviewed drafts of the article, and approved the final draft.

## Data Availability

Raw data and code are in the Supplemental Files. The data is also available at GitHub and Zenodo: https://github.com/KingArthur000/Electrical-Fault-detection-and-classification.git.

TangZwei. (2023). TangZwei/pso: v 1.0 (RELEASE). Zenodo. https://doi.org/10.5281/zenodo.7543703.

## Supplemental Information

Supplemental information for this article can be found online at http://dx.doi.org/10.7717/peerj-cs.1213#supplemental-information.

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
