# Peer review of "Research on fault location algorithm of TPSS based on PSOA"

_PeerJ Computer Science, doi:10.7717/peerj-cs.1213_

## Round 0.1 · original submission · Major Revisions

Dear Author
Thank you for your submission, Your paper has been reviewed by the experts in the field and you will see that they are of the view that your proposed research is of good worth, but it needs some improvements. Therefore, you are requested to revise your paper according to technical comments and re-submit.

Moreover, as your research deals with traction power supply system (TPSS) protection technology in order to ensure the safe operation of urban rail transit, which is a very interesting topic, I suggest you change your paper title accordingly so that it could correctly reflect the proposed research. Thank you

Reviewer 1 ·

Basic reporting

In this paper, the author develops a fault location algorithm and an evaluation of multi-point monitoring data. The fault location algorithm is based on particle swarm optimization algorithm (PSOA) based on the dynamic characteristics of the TPSS. And the evaluation is based on fuzzy processing of the average value of polarization potential forward deviation and multi-attribute decision-making. In conjunction with the actual project, the model is trained using measured field data; the experiment achieves relatively satisfactory results. The manuscript can be accepted after the following problems are solved.

Experimental design

In the second paragraph of the introduction, the author introduces that a gratifying progress is being made on DC traction power supply return systems due to the rapid development of urban rail transit, but then the author introduces some protection principles and measures. Therefore, the expressions in this section are inconsistent. There are some problems with the author's writing in terms of logic. It is advised that the author recheck and correct the writing.

Validity of the findings

In the related works, the author presents the work of previous authors in turn and there is a causal relationship between references [9] and [10]. But by analysis, the causal relationship is not appropriate. The author needs to analyze this part carefully and make adjustments.

Additional comments

It is very important to have a clear hierarchy of articles. On the contrary, no hierarchy in the paper makes the structure confusing and the logic unclear. For example, in Section 3.2, the author states the principles of TPSSOA in one paragraph, making the logic unclear. The author ought to display it in order.

In Section 3.2, the author introduces the principles and specific steps of TPSSOA. This part aims to emphasize the improved method: TPSSOA proposed by this paper compared with PSOA the Section 3.1, so the author is recommended to change the title of Section 3.2 to “Specific scheme of TPSSOA”.
In order to make the article more readable, it is recommended that the author try to keep the tables and text descriptions together. If the tables and textual explanations are too scattered, it does not help the reader understand the tables' essence and read the experimental results. In Section 4, it may be better if the author could keep the tables and text descriptions together.
In Section the results are shown in Figure 4 and Figure 5, but there is no more introduction to the two figures. The author should first explain what the two figures are to show to make readers have an overall understanding.
In Section 4, the author illustrates the results of Figure 4 and Figure 5 to compare the two algorithms: IPSO and TPSSOA. Because there are only two algorithms, the author’s statements “fastest” and “slowest” are wrong. It is better to modify it into the correct form.

Reviewer 2 ·

Basic reporting

This paper proposes a fault location algorithm based on particle swarm optimization algorithm (PSOA) based on the dynamic characteristics of the TPSS and an evaluation of multi-point monitoring data based on fuzzy processing of the average value of polarization potential forward deviation and multi-attribute decision-making. The results show that TPSSOA can achieve optimal discharge current control. However, the common problems are as follows:
1. In a scientific paper, one must pay attention to the accuracy of the presentation and it is clear that the author has some problems in this respect. For example, in the introduction, the author states that DC traction is widely used in urban rail transit systems due to its characteristics and lists three points, but there is some ambiguity; the readers cannot judge whether the author intends to introduce the application or the characteristics. Please restate your opinion more clearly.
2. The author's writing is not logically coherent and has no transitions before or after. For example, in the introduction, the author states the comprehensive measures discussed by Chen et al. and the existing problems without any transitions making the paper lack connection. The author had better pay attention to the transitions while writing.
3. The author's handling of the abbreviated form seems inadequate. For example, in Section 3.1, there is no explanation when mentioning PSO and PSOA and readers have no idea what they refer to. I suggest the author explain the abbreviated forms in the paper, especially where they first appear.
4. In Section 3.1, the author introduces the schematic diagram of DG, explains it in the form of equations, states the variables in detail, the scholars' efforts, and the article's improvement in one paragraph. It makes this part redundant and difficult to read. It would be better if the author could state them separately.
5. The logic of the first paragraph of Section 4 does not make sense, and the grammatical expression is incorrect. You should probably consider the intention of this paragraph and adjust.
6. There are many grammatical and punctuation errors in the writing, so please improve them.

Experimental design

See the above comments.

Validity of the findings

See the above comments.

Additional comments

See the above comments.

---

## Round 0.2 · accepted · Accept

Thank you for your fine contribution to our esteemed journal. Good luck!

Reviewer 1 ·

Basic reporting

They are done.

Experimental design

They are done. It is very good.

Validity of the findings

They are done. It is very good.

Additional comments

No comment.

Reviewer 2 ·

Basic reporting

My previous comments have been addressed.

Experimental design

My previous comments have been addressed.

Validity of the findings

My previous comments have been addressed.

Additional comments

My previous comments have been addressed.